# Substrates of the Human Brain Proton-Organic Cation Antiporter and Comparison with Organic Cation Transporter 1 Activities

**DOI:** 10.3390/ijms23158430

**Published:** 2022-07-29

**Authors:** David A. Doetsch, Salim Ansari, Ole Jensen, Lukas Gebauer, Christof Dücker, Jürgen Brockmöller, Alexandra Sachkova

**Affiliations:** Institute of Clinical Pharmacology, University Medical Center, Georg-August University, Robert-Koch-Str. 40, 37075 Göttingen, Germany; david.doetsch@stud.uni-goettingen.de (D.A.D.); salim.ansari@med.uni-goettingen.de (S.A.); ole.jensen@med.uni-goettingen.de (O.J.); lukas.gebauer@med.uni-goettingen.de (L.G.); christof.duecker@med.uni-goettingen.de (C.D.); alexandra.sachkova@med.uni-goettingen.de (A.S.)

**Keywords:** organic cation transporter, proton-organic cation antiporter, hCMEC/D3 cells, blood–brain barrier, orphan transporter

## Abstract

Many organic cations (OCs) may be transported through membranes by a genetically still uncharacterized proton-organic cation (H + OC) antiporter. Here, we characterized an extended substrate spectrum of this antiporter. We studied the uptake of 72 drugs in hCMEC/D3 cells as a model of the human blood–brain barrier. All 72 drugs were tested with exchange transport assays and the transport of 26 of the drugs was studied in more detail concerning concentration-dependent uptake and susceptibility to specific inhibitors. According to exchange transport assays, 37 (51%) drugs were good substrates of the H + OC antiporter. From 26 drugs characterized in more detail, 23 were consistently identified as substrates of the H + OC antiporter in six different assays and transport kinetic constants could be identified with intrinsic clearances between 0.2 (ephedrine) and 201 (imipramine) mL × minute^−1^ × g protein^−1^. Excellent substrates of the H + OC antiporter were no substrates of organic cation transporter OCT1 and vice versa. Good substrates of the H + OC antiporter were more hydrophobic and had a lower topological polar surface area than non-substrates or OCT1 substrates. These data and further research on the H + OC antiporter may result in a better understanding of pharmacokinetics, drug–drug interactions and variations in pharmacokinetics.

## 1. Introduction

The membrane transport of most organic cations is not well understood. Roughly estimated, one-third of all low-molecular weight drugs and other substances of natural or synthetic origin are organic cations. As such, they are majorly or almost completely positively charged at the typical pH in the human body. Particularly when they are more hydrophilic, such positively charged substances may hardly cross biological membranes by diffusion. More lipophilic substances may not only pass membranes by nonionic diffusion but also with these substances, influx and efflux transport may be significantly accelerated by membrane transporters. For several organic cations, organic cation transporters such as OCT1, OCT2, OCT3, OCTN1, OCTN2, THTR1, THTR2, MATE-1, MATE-2K and several others have been identified as relevant transporters [1,2,3]. However, these known transporters do not contribute in a relevant manner to the membrane transport of many cationic drugs. For many of these drugs, the H + OC antiporter may mediate the transport. Among these drugs are several first-generation antihistaminergic drugs, nicotine, varenicline, oxycodone and clonidine [4]. This H + OC antiporter was first described more than 20 years ago [5,6] and has a broad cell and tissue expression. In particular, its expression in the blood–brain barrier was studied in humans or human cell lines, but the H + OC antiporter is also expressed in the kidneys, liver, gut, lung and placenta [7,8,9,10,11,12]. Many studies on the H + OC antiporter deal particularly with the H + OC antiporter activity in the hCMEC/D3 cell line, which was derived by the lentivirus transfection of human cerebral endothelial cells. This cell line stably expresses many biochemical features of the endothelial cells at the blood–brain barrier [13,14].

The identification of the relevant membrane transporters for any drug and other substances is most relevant for understanding pharmacokinetics, drug–drug interactions and the effects of inherited genomic variation in pharmacokinetics. A clear identification of the relevant membrane transporters for a given substrate is, currently, mostly achieved by using model cell lines with overexpression or knock-outs. These techniques, however, cannot yet be applied for the study of the H + OC antiporter, since the gene or the genes coding for the H + OC antiporter have not been discovered yet. Therefore, we have to rely on functional transport parameters, which are characteristic for this transporter. Interestingly, the H + OC antiporter cannot only exchange protons with organic cations but also organic cations from both sides of the cell membrane. Measuring this exchange activity (also termed as trans-stimulation in this research field) is an important tool for identifying substrates of the H + OC antiporter. To perform this, cells are typically preloaded with probe substrates such as diphenhydramine. When adding another substance after preloading with a probe substrate, the rapid washout of the probe substrate indicates carrier-mediated organic cation exchange [15,16]. In addition, the H + OC antiporter is characterized by typical inhibitors such as imipramine, diphenhydramine, clonidine or memantine [17], but to our best knowledge, there is no inhibitor specific for the H + OC antiporter. Many of these inhibitors may also inhibit, for instance, OCT1 [2] and inhibitors such as verapamil or quinidine are also very pleiotropic and, therefore, less useful for transporter specification. A third line of evidence comes from disturbing the pH gradient by protonophores such as carbonyl cyanide-p-trifluoromethoxyphenylhydrazone (FCCP) [18].

Using several criteria including those described above, thus far, about 60 substances have been identified as substrates of the H + OC antiporter. A more comprehensive knowledge of the substrate spectrum of the H + OC antiporter may help better understand the pharmacokinetics of organic cations, interindividual variation in pharmacokinetics of organic cations and, for instance, drug–drug interactions at the blood–brain barrier. It may also help to elucidate the genetic identity of the H + OC antiporter. Thus, in this study, we screened 72 medically relevant and mostly positively charged drugs as substrates of the H + OC antiporter (Table 1). Substrates were further preselected by a molecular weight below 600 and mostly, but not exclusively, we tested hydrophilic substrates with a logD7.4 value below 2. Of the 72 substrates tested here, to our knowledge only 3 substrates (clozapine, oxycodone and methylphenidate) were earlier identified as substrates of the H + OC antiporter.

In addition, we tested whether or not substrates of the H + OC antiporter might also be substrates of the organic cation transporter 1 (OCT1). This was motivated by our observation that the typical substrates of the H + OC antiporter were not or only poor substrates of OCT1, indicating specific differences in the pharmacophores of the H + OC antiporter and OCT1.

## 2. Results

### 2.1. Testing 26 Possible Substrates by Applying 6 Criteria for H + OC Antiporter Substrates

In the first step, we performed a detailed analysis of 26 drugs concerning typical parameters indicating transportation via the H + OC antiporter, namely the (a) antiport with two prior established H + OC antiporter substrates (diphenhydramine, 5,6-Methylenedioxy-2-aminoindane), (b) inhibition by typical inhibitors of the H + OC antiporter (imipramine, diphenhydramine and memantine) and (c) modulation (reduction expected) of the transport by a disruptor of the cell-membrane proton gradient (FCCP). The 26 drugs were selected from medically relevant drugs by chemical criteria typical for organic cation (solute carrier, SLC) transporter substrates (pka > 7.4, a molecular weight < 600, and a LogD at pH 7.4 < 2). The ratios indicating transport or inhibition of transport are summarized in Table 2.

Most tested substances, except for morphine and pyrazinamide, were subject to extensive exchange transport during the 2-min incubation, as indicated by ratios significantly below unity when using DPH or MDAI as antiporter substrates (Table 2).

Also in the inhibition experiments, morphine and pyrazinamide were the only two substances where the uptake of which was not inhibited by the three inhibitors used here (Table 2, Figure 1B,C). As indicated in Figure 1, inhibition by DPH and MDAI showed a strong correlation and showed that imipramine was generally a stronger inhibitor while memantine was a weaker inhibitor, corresponding also with KM values of 20 µM and 29 µM identified in this study for imipramine and memantine, respectively.

As indicated by the summarizing view in the radar plot, all substances except for morphine and pyrazinamide were identified as possible substrates of the H + OC antiporter. The latter two were also not affected at all by the protonophore FCCP (Figure 1D, upper right square). However, the effects of FCCP correlated generally less well with inhibition and trans-stimulation parameters, probably due to other off-target interactions of FCCP with the transport process, such as alterations in intracellular energy metabolism and/or other effects on activity, e.g., by binding to the transporter protein.

We then measured the concentration-dependent uptake in hCMEC/D3 cells and calculated the transport kinetic parameters (K_M_, V_max_, intrinsic clearance (Cl_int_)). The concentration–uptake curves are shown in Figure 2. As observed, for 24 of the 26 organic cations tested, there was a saturable concentration-dependent uptake into hCMEC/D3 cells.

Transport kinetic constants are summarized in Table 3. Bupropion, citalopram, imipramine, duloxetine and memantine were transported with highest affinity (K_M_ below 30 µM). Imipramine, duloxetine and clozapine had the highest intrinsic clearance (above 100 mL/min/g protein). Ephedrine stood out with a very low maximal transport rate. However, with the relatively low K_M_, the H + OC antiporter might still be relevant in vivo.

In conclusion, of these experiments with 26 possible substrates of the H + OC antiporter, only 2 substrates, pyrazinamide and morphine, did clearly not fulfill the criteria of H + OC antiporter substrate (see also Figure 1). For a third substrate, metoprolol, no saturated transport could be seen in concentration-dependent uptake measurements, indicating that metoprolol may mainly be taken up by simple (not transporter mediated) diffusion. Although metoprolol might be a substrate of the H + OC antiporter according to the trans-stimulation and inhibition criteria (Table 2), we did not classify metoprolol as an H + OC antiporter substrate because there was no saturated transport.

### 2.2. Testing Further 46 Possible Substrates by Antiport (Trans-Stimulation) Assays

From the measurements shown above but also from published data we concluded, antiport and inhibition measurements are highly correlated and, therefore, redundant for characterizing a substance as a substrate of the H + OC antiporter. Therefore, we tested another set of 46 substances only with the antiport assays using diphenhydramine and MDAI (Table 1 and Table 4). As seen in Figure 3, which shows the correlation between the two trans-stimulation assays with prototypic exchange substrates diphenhydramine and MDAI, both assays correlated well.

While there was no visually apparent cut-off, the 72 substances tested here could be differentiated into 53 (74%) substrates of both exchange partners (using any ratio below 1 as indicative for antiport) and 37 (51%) good substrates (using a ratio below 0.7 as indicative for good antiport).

### 2.3. Relationship between H + OC Antiporter Substrates and OCT1 Substrates

Another question of this study was whether there may be an overlap or distinction of substrates of the H + OC antiporter and substrates of the well-characterized organic cation transporter OCT1. As illustrated in Figure 4, when using diphenhydramine as a probe substrate, only four of the substances tested were good substrates of both the H + OC antiporter and OCT1 (dobutamine, milnacipran, desmethylvenlafaxine and meptazinol). When using MDAI as antiporter probe substrate (right panel in Figure 4), there was almost a mutual exclusion concerning good antiporter substrates and good OCT1 substrates. All substances with their diphenhydramine and MDAI trans-stimulation ratios and their OCT1 uptake ratios are provided in Table 4.

### 2.4. Chemical Characteristics of H + OC Antiporter Substrates

Next, we compared if non-substrates, poor substrates and good substrates of the H + OC antiporter might differ in some of the most relevant chemical properties and we also compared this with the substrate patterns of OCT1. As shown in Figure 5, neither H + OC antiporter substrates nor OCT1 substrates can be distinguished by molecular weight, but substances with a molecular weight above 400 Dalton are rarely substrates of the antiporter or of OCT1. Interestingly, the smaller the topological polar surface area (TPSA), the higher the likelihood it is a substrate of the H + OC antiporter. This is also compatible with the fact that a TPSA below 70 Å is a typical feature of substances passing well through the blood–brain barrier [19]. Only one substrate, trimethoprim, had a TPSA above 100. Trimethoprim may not really be a good H + OC antiporter substrate, since it was classified as such only according to the diphenhydramine antiport assay results but not with MDAI (Table 4). Correspondingly, substrates of the H + OC antiporter were significantly more hydrophobic compared with non-substrates or with substrates of OCT1 (Figure 5). While OCT1 substrates only rarely have a logD value above 1.5, several H + OC antiporter substrates have logD values around and even above 2. High lipophilicity may constitute a general upper limit of SLC influx transporters simply due to the then prevailing passive and non-ionic diffusion of very lipophilic substrates. However, as shown here, this limit concerning lipophilicity depends on the transporter (Figure 5).

Finally, because the question about the genetic identity of the H + OC antiporter is still unanswered, we looked if selected typical substrates of the H + OC antiporter might be substrates of one of the four well-characterized proton-dependent membrane transporters with antiport activity, namely the two OCTN transporters and the two high-affinity thiamine transporters, THTR1 and THTR2. Although there were indeed minor transport activities found for several of these other antiporters (Figure 6), the ratio of the cell uptake into overexpressing cells over empty-vector transfected cells almost always was below 2, a cutoff frequently used to define a medically relevant transport activity.

## 3. Discussion

Since the protein conferring H + OC antiport activity is still not identified, there is no completely unequivocal assay for identifying H + OC antiporter substrates. Moreover, currently, we cannot exclude that the apparently antiporter-mediated uptake of more than 40 substrates that are not mediated by one but by several proteins. Nevertheless, from the first set of extensively tested 26 substrates, 23 consistently turned out to be H + OC substrates according to trans-stimulation, inhibition, pH gradient dependence and concentration-dependent (saturated) transport. From the entire set of 72 drugs, based on a cutoff of 0.7 in the diphenhydramine antiport assay, 42 turned out to be good substrates (Table 4). We have preselected our substrates for more hydrophilic cationic substances with a molecular weight below 600 Dalton, but it is nevertheless remarkable that for this large subset of drugs, about 50% were transported by the H + OC antiporter.

Here, we specifically looked at the correlation between transport via the organic cation uniporter OCT1 and the H + OC antiporter. Although not being absolutely mutually exclusive, the correlation showed that apparently very good OCT1 substrates are often not H + OC antiporter substrates and vice versa (Figure 4). As known, OCT1 substrates are rarely if ever highly lipophilic [1,21], but as illustrated in Figure 5, lipophilicity alone does not explain the difference between OCT1 and H + OC antiporter substrates. A simple argument why lipophilic substances are not substrates of OCT1 may be that passive diffusion in either directions may mask the effect of transporter activities. In this respect, the so-called trans-stimulation assay (data for all 72 substances tested in Table 4) may be less dependent on lipophilicity. Indeed, lipophilic substances such as amitriptyline or its metabolite nortriptyline, which are not OCT1 substrates [22], had a reasonable intrinsic clearance as measured by influx into the hCMEC/D3 cells (Table 3). These data also indicate that overrunning carrier-mediated transports by simple diffusion in both directions cannot be the only explanation why OCT1 substrates are typically not highly lipophilic.

Since one may argue that primarily OCT2 and OCT3, but not OCT1, are the extrahepatic transporters, we also looked at that correlation between transport by the antiporter (DPH antiport and MDAI antiport; see Table 4) and OCT2 and OCT3 transport ratios in a subset of 22 of the substances tested here (detailed data not shown). However, the picture with OCT2 and OCT3 was essentially the same as with OCT1 (Figure 4). Thus, good substrates of the H + OC antiporter were not or only poor substrates of OCT2 or OCT3.

While the majority of ABC and SLC transporters has currently been identified and characterized concerning at least of some of their substrates [3,23], the genetic identity of the H + OC antiporter has still not been disclosed. One approach to this question might be testing the substrates of the H + OC antiporter as substrates of well-defined SLC transporters. This has been studied earlier by several authors using specific chemical inhibitors and siRNA-based knockdown [24,25]. Here, we also studied the transport of prototypic H + OC antiporter substrates (clonidine, cocaine, diphenhydramine, MDMA, memantine, oxycodone, varenicline and MDAI) as substrates of two human high-affinity thiamine transporters and the two OCTN transporters, all four of which are expressed at a moderate level in the hCMEC/D3 cells [26]. However, based on the ratios of cell uptake with overexpressing cells over the uptake in empty-vector transfected cells, there was either no transport or only low transport ratios (Figure 6). Of course, one might test all the 72 substances characterized here with the more than 12 organic cation transporters known at present, but this was beyond what was achievable in our present project.

Since we contrasted here the activity of the primarily hepatically expressed OCT1 to the activity of the H + OC antiporter, one may ask whether OCT1 contributed to transports in the hCMEC/D3 cells. Indeed, OCT1 may be expressed in the hCMEC/D3 cells [26] but only at quite low concentrations. Moreover, as demonstrated here (Figure 4, Table 4), OCT1 is mostly not a good transporter for H + OC antiporter substrates.

Currently, we cannot even answer whether the H + OC antiporter is one protein or many, so we are at a similar stage as our colleagues in studying drug metabolism who asked 50 years ago, “Cytochrome P-450 of liver microsomes—one pigment or many” [27]. However, the generally rather high correlation between the different assays used here (Figure 1) might indicate that it is, indeed, mostly one protein, because with different transport proteins, more variability in the results of the different assays between different substrates might exist.

The medical impact of the finding, that about 50% of the drugs tested here are substrates of the H + OC antiporter, may be significant. For instance, all substrates of the H + OC antiporter may exhibit significant drug–drug interactions between each other when passing through the blood–brain barrier and considering that, in many fields of medicine, polymedication with five or more drugs is widely practiced, and drug concentrations achieved in the brain may be very different in polymedication compared with monotherapy. However, of course, this aspect has to be investigated in further experimental and clinical studies.

Another quite interesting fact is that the pharmacokinetics and pharmacodynamics of several substrates of the H + OC antiporter show extremely large variations between individuals. At the very least, there are a number of examples for that. For instance, varenicline, an H + OC antiporter substrate [28], has blood concentrations varying by more than 100-fold in published studies [29]. Similarly, oxycodone is a well-established substrate of the H + OC antiporter [13] and pharmacokinetics and pharmacodynamics of oxycodone also showed an extreme interindividual variation [30]. Clonidine may serve as a third example of a H + OC antiporter substrate with highly variable pharmacokinetics and pharmacodynamics [12]. Notably, the variation of these three drug examples is not explained by known variations in genetically polymorphic drug metabolizing enzymes such as CYP2D6 since these enzymes play no or only a minor role here. Thus, it would be interesting to know whether there is significant genomic variation in the gene coding for the H + OC antiporter or if the activity of the antiporter is significantly modulated by other endogenous substances. However, since we do not know the gene yet, we cannot assess whether this variation is due to acquired or inherited factors.

In conclusion, depending on the functional test and the cutoff value used, up to 50% of drugs tested here that have a pKa > 7.4 and a molecular weight below 500 Dalton may be substrates of the proton-organic cation antiporter. Earlier, more than 60 substrates of the H + OC antiporter have been identified [31], and this number is significantly increased by the substrates identified in this publication. This knowledge may allow designing drugs with better blood–brain barrier penetration [32]. The H + OC antiporter is also expressed in many other tissues. It may be that this H + OC antiporter expressed in many other tissues is coded by the same gene(s) as the antiporter(s) expressed in the hCMEC/D3 cell line used in our experiments. In that case, our data may allow a better understanding of pharmacokinetics, drug–drug interactions and variations in pharmacokinetics and drug effects. Finally, the study illustrates that further efforts to identify the gene(s) coding for the H + OC antiporter(s) are very important.

## 4. Materials and Methods

All transport measurements were performed in hCMEC/D3 cells [14], which were cultured in an RPMI 1640 medium (Sigma-Aldrich, Taufkirchen, Germany) supplemented with 10% (*v*/*v*) fetal bovine serum, 1% penicillin/ streptomycin and 1 ng/mL bFGF as described earlier [16]. The cells were cultured for a maximum of 30 passages in rat collagen I (R&D System, Minneapolis, MN, USA)-coated 75 mm flasks at 37 °C at 95% relative humidity and 5% CO2. Transport assays were performed in collagen I-coated 12-well plates, in which 0.3 million cells/well were seeded and grown for 72 h to reach more than 90% confluence. All compounds used were purchased with purities > 95% from renowned manufacturers (Sigma-Aldrich, Taufkirchen, Germany; Toronto Research Chemicals, Toronto, ON, Canada; Santa Cruz Biotechnology, Darmstadt, Germany; Tocris, Bioscience, Bristol, UK).

For concentration-dependent uptake, cells were washed with pre-warmed HBSS+ (Hanks’ balanced salt solution supplemented with 10 µM HEPES, pH 7.4) and then incubated with increasing concentrations (0.1–1000 µM) of pre-warmed substrate solution prepared in HBSS+ for two minutes at 37 °C. The uptake was stopped by adding ice-cold HBSS+, and cells were immediately washed twice with ice-cold HBSS+. After washing, the cells were lysed with a lysis buffer (80% (*v*/*v*) acetonitrile, LGC Standards GmbH, Wesel, Germany) containing an individual internal standard (Table 5). For lysis, cells were incubated for 10 min on a shaking device at a slow shaking rate. Intracellular substrate accumulation was measured by high-performance liquid chromatography-coupled mass spectrometry (HPLC-MS/MS).

For protein measurements from each cell culture plate, two wells were not used for transport assays. These wells were also treated with a lysis buffer for ten minutes. These lysates were then transferred to a 96-cell plate and protein concentrations were measured using a bicinchoninic acid assay [33].

For trans-stimulation experiments, cells were preloaded with the antiport probe drugs (diphenhydramine or MDAI, 1 µM final concentration) for 30 min and then quickly washed once with pre-warmed HBSS+ and then incubated with each substrate (250 µM final concentration) for two minutes. The exchange transport was stopped by adding ice-cold HBSS+ and cells were further processed as described above for analysis. Results were expressed as the ratio of the intracellular antiport probe drug concentration by adding a substrate for 2 min over the concentration without adding a substrate. Any ratio significantly below unity indicates an antiport.

Inhibition is also often termed cis-inhibition in the study of the H + OC antiporter acknowledging that the substrate and inhibitor act from the same side of the cell. For these inhibition experiments, the known inhibitors memantine [28], diphenhydramine and imipramine [17] were used at 250 µM and substrates were used at a 3 µM final concentration prepared in HBSS+. Both substrates and inhibitors were pre-mixed and pre-warmed before the incubation for two minutes at 37 °C. The uptake was stopped by adding ice-cold HBSS+ and the cells were further processed as described above for analysis of the intracellular substrate accumulation. Results were expressed as the ratio of the intracellular concentration of the probe drug with inhibition over its intracellular concentration without inhibition.

Carbonyl cyanide-p-trifluoromethoxyphenylhydrazone (FCCP) was used to investigate proton gradient dependency for uptake. Substrates were incubated for two minutes simultaneously either in the presence of FCCP (25 µM final concentration) or in the absence. Further steps were performed as described above. The final concentration of ethanol (solvent of the 10 mM FCCP stock solution) did not exceed 0.2% (*v*/*v*) per reaction. Results were expressed as the ratio of intracellular substrate concentration with FCCP added over the intracellular concentration without FCCP.

Intracellular concentrations of probe substrates and substrates-in-question were performed using HPLC-MS/MS analyses as essentially as described earlier [1,2,16]. Briefly, the cell lysate was centrifuged at 13,000 rpm in a tabletop centrifuge for 15 min to pellet potential cell debris. Supernatant was then transferred to a deep-well collection plate and samples were evaporated at 37 °C under constant nitrogen flow until dry and subsequently reconstituted in 0.1% (*v*/*v*) formic acid. HPLC was conducted using a Shimadzu Nexera HPLC system with a LC-30AD pump, a SIL-30AC autosampler, a CTO-20AC column oven and a CBM-20A controller (all Shimadzu, Kyoto, Japan). Liquid chromatography was performed on a Brownlee SPP RP-Amide column (4.6 × 100 mm inner dimension with 2.7 μm particle size; PerkinElmer, Waltham, MA, USA) with a C18 pre-column at an oven temperature of 40 °C. The aqueous mobile phase contained 0.1% (*v*/*v*) formic acid and either 3, 8 or 20% (*v*/*v*) organic additive (acetonitrile:methanol 6:1 (*v*/*v*), LGC Standards GmbH, Wesel, Germany). The separation of samples (injection volume: 10 µL) was carried out with a flow rate of 300 µL/min. An API 4000 tandem mass spectrometer (AB SCIEX, Darmstadt, Germany) was used in a multiple reaction monitoring (MRM) mode for the detection of substances after positive ionization. Internal standards and masses are provided in Table 5. The Analyst software, version 1.6.2, from AB SCIEX was used for the quantification. For the quantification of intracellular concentrations, a calibration curve with known concentrations consisting of at least seven data points was measured for each substance and experiment.

Data analysis: Here, we assume that any influx or efflux transport is the sum of at least two types of transport, passive diffusion and carrier-mediated transport [7]. Since the gene encoding the H+/OC antiporter is still unidentified, we cannot perform uptake experiments overexpressing the H+/OC antiporter as performed with cell lines with and without overexpression of well-identified transporters. Therefore, the net transport mediated by the overexpressed transporters could not be calculated by subtracting the cellular uptake with an empty-vector cell line. However, assuming that the increase in concentration-dependent cell uptake between the two highest substrate concentrations is mostly due to passive diffusion, we subtracted the concentrations corresponding to this slope from the measured intracellular concentrations and used the corresponding curve for the determination of the transport kinetic constants, as described earlier [16].

K_M_ and V_max_ values with corresponding standard errors were calculated from at least three independent experiments by nonlinear regression analysis using the Michaelis–Menten equation (v = (V_max_ × [S])/(K_M_ + [S]) and using the least squares fit method in GraphPad software. The intrinsic clearance Clint was calculated as the ratio of Vmax over K_M_.

The uptake ratio for cis-inhibition, trans-stimulation and proton-dependent uptake experiments was calculated relative to the respective control experiment as uptake ratio = intracellular concentration/intracellular concentration in the control. The differences between the test and the control experiment in cis-inhibition, trans-stimulation and proton-dependent uptake experiments were tested for statistical significance using the Student’s *t*-test. Transport kinetic parameters were calculated in the GraphPad Prism software version 5.01 for Windows (GraphPad Software, La Jolla, CA, USA). The chemical features of all substrates were determined using Instant JChem (ChemAxon, Budapest, Hungary) processing the individual isomeric SMILES from PubChem [34].

## Figures and Tables

**Figure 1 ijms-23-08430-f001:**
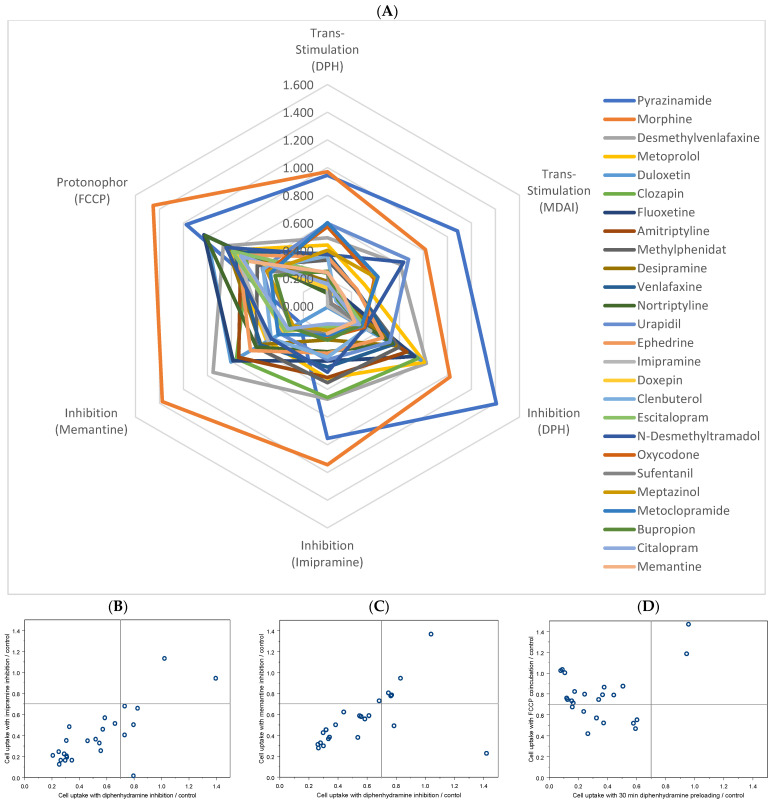
Correlation between different assays indicative for possible H + OC antiporter substrates. (**A**) Radar graphic with small values (lines near the center) strongly indicate that a substance is a substrate of the H + OC antiporter. Only two substances on the outer circle, morphine and pyrazinamide, are clearly not H + OC antiporter substrates. The roughly concentric circles in this graph should illustrate a good correspondence between the 5 different indicators of H + OC antiporter substrates (data for each substance and assay in Table 2). (**B**) Correlation between diphenhydramine and imipramine inhibition, (**C**) correlation between diphenhydramine and memantine inhibition and (**D**) correlation between diphenhydramine trans stimulation ratio and FFCP inhibition effect.

**Figure 2 ijms-23-08430-f002:**
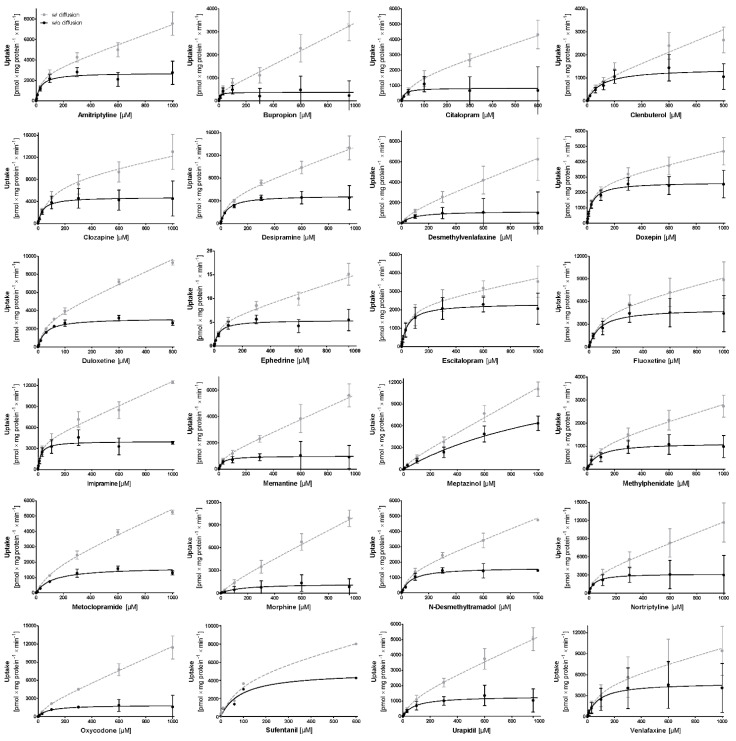
Concentration-dependent uptake of substrates tested as possible H + OC antiporter substrates. Grey lines show the uptake rates as measured and black lines show the uptake ratios after a subtraction of the most likely diffusion-related component. Error bars indicate a standard error of a mean of at least 3 separate incubations. For organizational reasons, sufentanil measurements could only be performed once. With metoprolol and pyrazinamide, only a linear concentration–uptake relationship was observed (not shown), disproving evidence related to carrier-mediated transport in the hCMEC/D3 cells.

**Figure 3 ijms-23-08430-f003:**
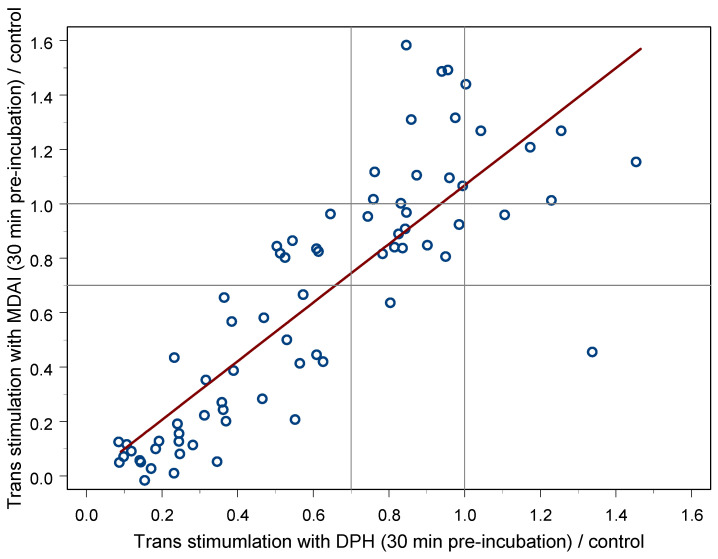
Correlation between the trans-stimulation (antiport) with diphenhydramine (DPH) and 5,6-Methylenedioxy-2-aminoindane (MDAI). Cells were pre-incubated for 30 min with either DPH or MDAI and the percentages refer to the percentage remaining within the cells 2 min after addition of the substrates. While there is no clearly apparent cutoff between substrates and non-substrates, all substances above 1 are per definition non-substrates and all substances below 0.7 may be considered as good substrates. Therefore, these reference lines are shown in the graph together with the linear regression line. As observed, the regression line is very similar to the line of identity.

**Figure 4 ijms-23-08430-f004:**
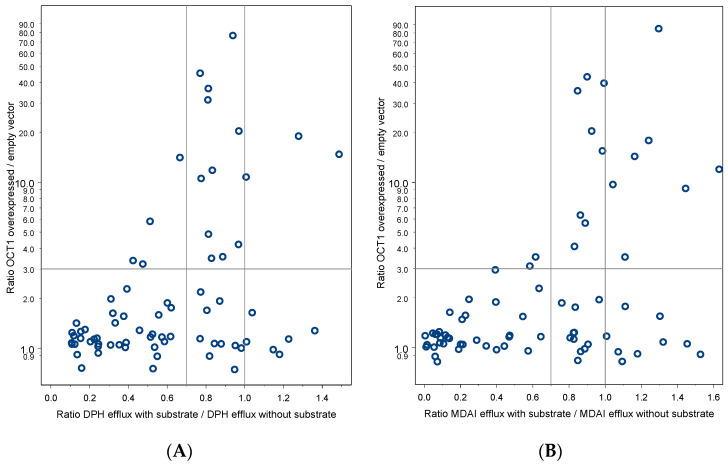
Relationship between H + OC antiport and transport by OCT1. (**A**) In the left figure, trans-stimulation activity with diphenhydramine is used as indicator of H + OC antiport and the ratio of cell uptake in OCT1 overexpressing HEK293 cells over empty-vector transfected cells was used as an indicator of OCT1 transport. (**B**) In the right figure, the corresponding data from trans-stimulation activity with MDAI were used as an indicator of the H + OC antiport. As observed, in the upper left, there are almost no substances indicating that being a good OCT1 substrate and being a good H + OC antiporter substrate are largely exclusive (specific substance data in Table 4).

**Figure 5 ijms-23-08430-f005:**
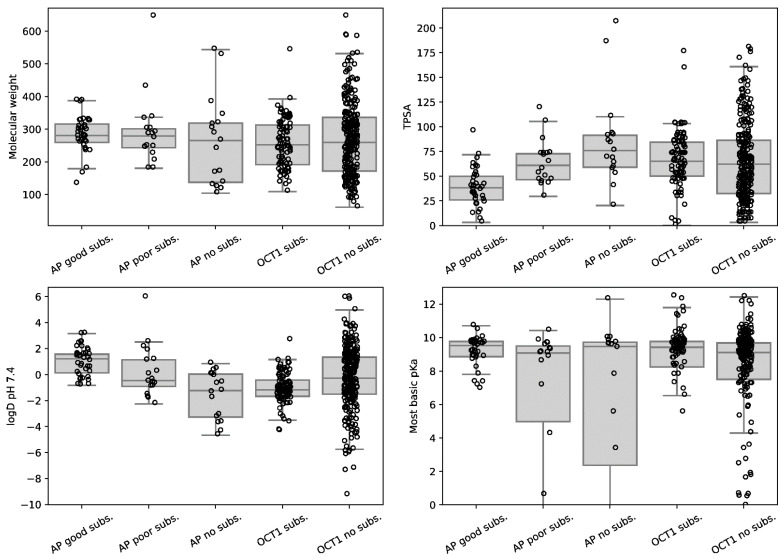
Chemical characteristics of good, poor and non-substrates of the H + OC antiporter and of OCT1. Substrates (subs.) and non-substrates of OCT1 were published earlier [1,20]. AP, H + OC antiporter; TPSA, topological polar surface area.

**Figure 6 ijms-23-08430-f006:**
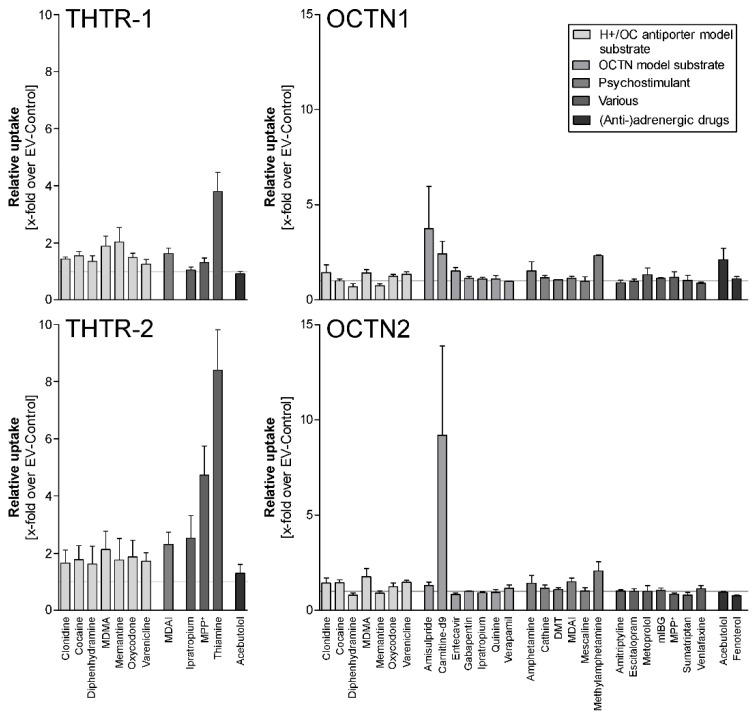
Prototypical substrates of the H + OC antiporter and some other drugs tested as substrates of the high-affinity thiamine transporters THTR-1 and THTR-2 and of OCTN1 and OCTN2. All bars indicate the relative uptake of transporter-overexpressing cells over empty-vector (EV) transfected cells performed in at least 3 replicates. The two thiamine transporters and OCTN1 are also antiporters, but at least compared with the transport rate of their typical substrates, they are not the relevant antiporters.

**Table 1 ijms-23-08430-t001:** Drugs studied as substrates of the H + OC antiporter and of OCT1.

Substance	MW	LogD at pH 7.4	TPSA	Net Charge at pH 7.4	Strongest Basic pKa
Amitriptyline	277.4	2.5	3.24	1	9.8
Bupropion	239.7	2.4	29.1	1	8.2
Citalopram	324.4	1.4	36.26	1	9.8
Clenbuterol	277.2	0.1	58.28	1	9.6
Clozapine	326.8	3.1	30.87	0	7.4
Desipramine	266.4	1.4	15.27	1	10
Desmethylvenlafaxine	263.4	0.9	43.7	1	9
Doxepin	279.4	1.5	12.47	1	9.8
Duloxetine	297.4	1.9	21.26	1	9.7
Ephedrine	165.2	−0.8	32.26	1	9.5
Escitalopram	324.4	1.4	36.26	1	9.8
Fluoxetine	309.3	1.8	21.26	1	9.8
Imipramine	280.4	2.5	6.48	1	9.2
Memantine	179.3	−0.8	26.02	1	10.7
Meptazinol	233.4	1.6	23.47	1	9.2
Methylphenidate	233.3	0.6	38.33	1	9.1
Metoclopramide	299.8	−0.3	67.59	1	9
Metoprolol	267.4	−0.5	50.72	1	9.7
Morphin	285.3	−0.6	52.93	1	9.1
N-Desmethyltramadol	249.4	−0.4	41.49	1	9.9
Nortriptyline	263.4	1.6	12.03	1	10.5
Oxycodone	315.4	−0.4	59	1	8.8
Pyrazinamide	123.1	−1.2	68.87	0	−0.6
Sufentanil	386.6	2.1	32.78	1	8.9
Urapidil	387.5	0.6	68.36	1	7.8
Venlafaxine	277.4	1.1	32.7	1	9.1
**Additional substrates tested by antiport (trans-stimulation) assays only**					
Acebutolol	336.4	−0.7	87.66	1	9.7
Acyclovir	225.2	−1.6	114.76	0	0.6
Amiodaron	645.3	6	42.68	1	9.1
Atropin	289.4	−0.4	49.77	1	9.4
Betaine	117.1	−3.7	40.13	0	14
Bisoprolol	325.4	0	59.95	1	9.7
Buflomedil	307.4	0.6	48.00	1	8.7
Choline	104.2	−4.7	20.23	1	14 *
Daunorubicin	527.5	0.9	185.8	1	10
Dihydrocodeine	301.4	−0.4	41.93	1	9.3
Dobutamine	301.4	1.2	72.72	1	9.3
Doxorubicin	543.5	0	206.1	1	10
Eletriptan	382.5	1.6	53.17	1	9.6
Ethambutol	204.3	−2.3	64.52	1	9.6
Etomidate	244.3	2.5	44.12	0	4.3
Fenoterol	303.4	0.3	92.95	1	9.6
Formoterol	344.4	0	90.82	1	9.8
Frovatriptan	243.3	−1.7	70.91	1	10.4
Hydrocodone	299.4	0.2	38.77	1	9.2
Hydromorphone	285.3	0	49.77	1	9.1
Ipratropium	332.5	−1.8	46.53	1	14 *
Isoniazid	137.1	−0.7	68.01	0	3.4
Ketamine	237.7	3.1	29.10	0	7.2
Labetalol	328.4	1.3	95.58	1	9.8
Levetiracetam	170.2	−0.6	63.40	0	−1.6
Meropenem	383.5	−4.4	110.2	0	9.4
Metformin	129.2	−3.7	91.49	1	12.3
Methylscopolamin	318.4	−3.3	59.06	1	14 *
Milnacipran	246.4	−0.9	46.33	1	9.8
Nortilidine	259.3	1.5	38.33	1	8.9
Ondansetron	293.4	2.1	39.82	0	7.3
Paraxanthine	180.2	0.2	67.23	0	−1.1
Paroxetin	329.4	0.8	39.72	1	9.8
Phenylephrine	167.2	−1.4	52.49	1	9.7
Pindolol	248.3	−0.5	57.28	1	9.7
Pirbuterol	240.3	−1.8	85.61	1	9.6
Piritramide	430.6	1.9	73.36	1	8.9
Propranolol	259.3	0.4	41.49	1	9.7
Ranitidin	314.4	0.4	83.58	1	7.8
Scopolamin	303.4	0.8	62.30	0	7
Theophyllin	180.2	−0.9	69.30	0	−0.8
Thiamine	265.4	−3.1	75.91	1	5.5
Tilidine	273.4	2.1	29.54	1	8.6
Tranylcypromine	133.2	−0.8	26.02	1	9.6
Trimethoprim	290.3	1.1	105.5	1	7.2
Zolmitriptan	287.4	−0.1	57.36	1	9.5

The upper 26 substances were characterized concerning transport kinetics, trans-stimulation with diphenhydramine and MDAI as exchange substrates, inhibition with diphenhydramine, imipramine and memantine and pH gradient dependency using FCCP. The lower 46 were tested with trans-stimulation assays only. MW, molecular weight; TPSA, topological polar surface area, also known as polar surface area, which is the surface sum of all polar atoms (mostly oxygen and nitrogen; * quaternary amines are always positively charged in human body fluids and were arbitrarily labelled with a pka of 14. The chemical features of all substrates were determined using Instant JChem (ChemAxon, Budapest, Hungary).

**Table 2 ijms-23-08430-t002:** Parameters tested as indicators of transport via the H + OC antiporter.

	Trans DPH *		Trans MDAI		Cis DPH		Cis Imip		Cis Mema		FCCP	
Substance	M	SD	M	SD	M	SD	M	SD	M	SD	M	SD
Amitriptyline	0.15	0.08	0.03	0.03	0.66	0.18	0.51	0.28	0.7	0.2	0.72	0.06
Bupropion	0.25	0.12	0.09	0.08	0.27	0.04	0.25	0.06	0.3	0.1	0.44	0.22
Citalopram	0.16	0.07	0.1	0.01	0.27	0.02	0.13	0.03	0.3	0	0.72	0.04
Clenbuterol	0.36	0.15	0.07	0.02	0.33	0.14	0.37	0.17	0.5	0.2	0.54	0.01
Clozapin	0.16	0.07	0.03	0.02	0.75	0.08	0.66	0.18	0.8	0.1	0.67	0.1
Desipramine	0.18	0.1	0.1	0.07	0.57	0.14	0.24	0.03	0.6	0.1	0.83	0.04
Desmethylvenlafaxine	0.49	0.08	0.61	0.34	0.82	0.23	0.67	0.26	1	0.1	0.87	0.14
Doxepin	0.13	0.06	0.1	0.01	0.33	0.03	0.15	0.03	0.4	0	0.76	0.05
Duloxetin	0.1	0.01	0.1	0.09	0.78	0.06	0.01	0	0.8	0.1	1	0.03
Ephedrine	0.36	0.03	0.24	0.16	0.46	0.08	0.34	0.08	0.6	0.3	0.74	0.16
Escitalopram	0.23	0.08	0.13	0.07	0.32	0.02	0.15	0.04	0.4	0.1	0.79	0.06
Fluoxetine	0.1	0.03	0.13	0.12	0.73	0.05	0.39	0.04	0.8	0	1.03	0.17
Imipramine	0.13	0.02	0	0	0.36	0.07	NM	NM	0.5	0.1	0.75	0.09
Memantine	0.24	0.01	0.14	0.06	0.23	0.11	0.2	0.13	NM	NM	0.64	0.12
Meptazinol	0.4	0.04	0.41	0.31	0.29	0.06	0.18	0.04	0.3	0	-	-
Methylphenidate	0.33	0.05	0.24	0.09	0.6	0.23	0.55	0.16	0.6	0.2	0.58	0.14
Metoclopramide	0.6	0.17	0.42	0.35	0.29	0.17	0.21	0.22	0.4	0.2	0.48	0.25
Metoprolol	0.44	0.09	0.31	0.1	0.81	-	0.52	-	0.5	-	0.81	-
Morphine	0.97	0.1	0.82	0.33	1.02	0.68	1.14	0.16	1.4	0.3	1.45	0.34
N-Desmethyltramadol	0.37	0.1	0.64	0.1	0.31	-	0.48	0	0.5	0	0.85	-
Nortriptylin	0.09	0.05	0.03	0.04	0.54	0.01	0.33	0.06	0.59	0.03	1.02	0
Oxycodone	0.57	0.17	0.39	0.07	0.31	0.01	0.23	0	0.29	0.07	0.51	0.04
Pyrazinamide	0.94	0.51	1.08	0.19	1.41	-	0.95	-	0.23	-	1.18	-
Sufentanil	0.24	0.07	0.03	0.05	0.29	0.02	0.21	0	-	-	-	-
Urapidil	0.6	0.03	0.68	0.42	0.52	0	0.39	0.09	0.4	0.13	0.54	0.13
Venlafaxine	0.38	0.07	0.24	0.04	0.55	0.2	0.44	0.25	0.56	0.32	0.77	0.1

* Trans DPH, trans-stimulation (antiport) with diphenhydramine (DPH) as probe substrate, given as mean (M) and standard deviation (SD); activity was measured as ratio of DPH remaining within the cells after 2 min incubation with each of the 26 substrates over control incubation without substrate; Trans MDAI, trans-stimulation with 5,6-Methylenedioxy-2-aminoindane as a probe substrate, and parameters and procedures were the same as with DPH; Cis DPH, inhibition of the influx of each of the 26 substrates by the known H + OC antiporter substrate diphenhydramine, data given as ratio of intracellular substrate concentration after 2 min of incubation with inhibitor over substrate concentration without inhibitor; Cis Imip, inhibition with imipramine, parameters as above; Cis Mem, inhibition with memantine, parameters as above; FCCP, effect of FCCP on influx transport given as mean (M) ratio of intracellular substrate concentration with FCCP co-incubation over substrate concentration without FCCP co-incubation. NM, not meaningful; -, not measured or not measured in replicates.

**Table 3 ijms-23-08430-t003:** Transport kinetic parameters of substrate uptake into hCMEC/D3 cells.

Substrate	K_M_ (µM)		*V*_max_ [pmol × mg Protein^−1^ × min^−1^]		Intrinsic Clearance [mL × Minute^−1^ × mg Protein^−1^]	
	M	SEM	M	SEM	M	SEM
Amitriptyline	285.6	91.9	8752	1023	30.6	13.4
Bupropion	6.6	14.5	374	139	56.7	145.6
Citalopram	12.9	27.6	838	372	65.0	167.8
Clenbuterol	51.1	32.4	1412	263	27.6	22.7
Clozapine	34.9	28.7	4784	814	137.1	136.0
Desipramine	50.0	25.6	4918	552	98.4	61.4
DM venlafaxine *	87.3	20.3	1173	671	13.4	38.9
Doxepin	36.3	15.7	2661	239	73.3	38.3
Duloxetine	27.7	5.6	3167	155	114.3	28.7
Ephedrine	33.5	17.2	5.5	0.6	0.2	0.1
Escitalopram	47.0	22.7	2356	259	50.1	29.7
Fluoxetine	80.2	57.0	5070	869	63.2	55.8
Imipramine	20.1	9.8	4031	392	200.5	117.3
Memantine	28.7	39.5	1002	276	34.9	57.7
Meptazinol	1461	894	15,759	6348	10.8	10.9
Methylphenidate	78.9	56.7	1142	197	14.5	12.9
Metoclopramide	117.9	36.2	1671	137	14.2	5.5
Metoprolol	Only linear component of uptake observed					
Morphine	153.4	283.2	1243	666	8.1	19.3
N-DM tramadol **	69.5	22.7	1646	126	23.7	9.5
Nortriptyline	38.1	47.3	3205	833	84.1	126.3
Oxycodone	71.4	73.9	1900	462	26.6	34.0
Pyrazinamide	Only linear component of uptake observed					
Sufentanil	35.8	22.3	2983	489	83.3	65.6
Urapidil	82.5	80.4	1305	309	15.8	19.2
Venlafaxine	84.4	115.5	4874	1630	57.7	98.3

Cell uptake was measured for 2 min with a wide range of substrate concentrations. K_M_ and V_max_ were determined by nonlinear regression analysis with the Michaelis–Menten equation after subtraction of the linear passive-diffusion-related component. * desmethylvenlafaxine; ** N-desmethyltramadol.

**Table 4 ijms-23-08430-t004:** H + OC antiporter substrates identified by trans-stimulation with Diphenhydramine and MDAI in relation to uptake via OCT1.

Substrate	DPH Antiport		MDAI Antiport		OCT1 Ratio
	Ratio	SD	Ratio	SD	
Nortriptyline	0.09	0.05	0.03	0.04	1.14
Duloxetin	0.10	0.01	0.10	0.09	1.20
Fluoxetin	0.10	0.03	0.13	0.12	1.10
Propranolol	0.12	0.04	0.08	0.07	1.10
Doxepin	0.13	0.06	0.10	0.01	1.30
Paroxetin	0.13	0.00	0.07	0.03	1.17
Imipramine	0.13	0.02	0.01	0.00	1.10
Amitriptyline	0.15	0.08	0.03	0.03	1.20
Clozapin	0.16	0.07	0.03	0.02	1.00
Citalopram	0.16	0.07	0.10	0.01	1.30
Desipramine	0.18	0.10	0.10	0.07	0.80
Eletriptan	0.21	0.05	0.46	0.09	1.20
Ondansetron	0.22	0.09	0.22	0.08	1.00
Escitalopram	0.24	0.08	0.13	0.07	1.20
Memantine	0.24	0.01	0.14	0.06	1.10
Sufentanil	0.24	0.07	0.03	0.05	1.10
Bupropion	0.25	0.12	0.09	0.08	0.90
Tranylcypromine	0.30	0.14	0.11	0.08	1.60
Bisoprolol	0.32	0.08	0.35	0.10	1.00
Methylphenidat	0.33	0.05	0.24	0.09	1.90
Ketamine	0.36	0.12	0.23	0.11	1.10
Ephedrine	0.36	0.03	0.24	0.16	1.50
Clenbuterol	0.36	0.15	0.07	0.02	1.10
N-Desmethyl tramadol	0.37	0.10	0.64	0.10	2.40
Venlafaxine	0.38	0.07	0.24	0.04	1.50
Labetalol	0.40	0.06	0.57	0.18	1.00
Meptazinol	0.40	0.04	0.41	0.31	3.20
Metoprolol	0.44	0.09	0.31	0.10	1.20
Amiodaron	0.49	0.16	0.82	0.13	1.20
Desmethylvenlafaxine	0.49	0.08	0.61	0.34	3.30
Dihydrocodeine	0.52	0.18	0.87	0.36	0.80
Milnacipran	0.53	0.20	0.84	0.03	6.30
Buflomedil	0.53	0.30	0.22	0.15	1.00
Nortilidine	0.53	0.00	0.52	0.00	1.62
Tilidine	0.54	0.33	0.80	0.19	1.30
Oxycodone	0.57	0.17	0.39	0.07	0.90
Pindolol	0.59	0.16	0.83	0.06	1.20
Trimethoprim	0.60	0.37	0.85	0.33	1.90
Urapidil	0.60	0.03	0.68	0.42	1.10
Metoclopramide	0.60	0.17	0.42	0.35	1.80
Scopolamin	0.63	0.22	0.44	0.04	1.10
Dobutamin	0.67	0.13	0.99	0.59	14.70
Phenylephrine	0.76	0.11	1.03	0.23	10.60
Hydromorphone	0.76	0.07	0.79	0.28	2.00
Ipratropium	0.77	0.33	0.93	0.18	41.90
Metformin	0.79	0.16	1.10	0.13	1.70
Piritramide	0.80	0.00	0.82	0.02	1.18
Atropin	0.80	0.42	0.64	0.05	3.60
Frovatriptan	0.83	0.44	0.88	0.14	33.10
Acebutolol	0.84	0.07	0.88	0.07	5.20
Thiamine	0.84	0.04	1.60	0.53	11.60
Methylscopolamine	0.84	0.34	1.02	0.20	39.20
Etomidate	0.85	0.15	0.87	0.14	0.94
Daunorubicin	0.86	0.07	1.29	0.24	1.10
Acyclovir	0.87	0.28	0.94	0.28	2.00
Formoterol	0.87	0.28	1.10	0.20	3.60
Theophyllin	0.90	0.30	0.86	0.34	1.00
Doxorubicin	0.94	0.26	1.46	0.51	1.10
Pyrazinamide	0.95	0.51	1.08	0.19	0.80
Pirbuterol	0.95	0.28	1.31	0.38	80.90
Isoniazid	0.96	0.09	1.51	0.42	0.94
Morphin	0.97	0.10	0.82	0.33	4.30
Betaine	0.98	0.17	1.09	0.39	1.00
Ethambutol	0.98	0.46	0.95	0.10	20.80
Zolmitriptan	1.03	0.58	1.45	0.60	10.00
Choline	1.04	0.15	1.27	0.27	1.50
Paraxanthine	1.13	0.22	0.93	0.19	1.00
Levetiracetam	1.17	0.28	1.19	0.24	1.00
Meropenem	1.20	0.10	1.00	0.06	1.10
Ranitidin	1.26	0.67	1.24	0.36	18.10
Hydrocodone	1.35	1.54	0.46	0.18	1.20
Fenoterol	1.47	0.41	1.16	0.23	13.90

Substrates are ordered with decreasing exchange activity against diphenhydramine. The diphenhydramine ratio refers to intracellular diphenhydramine after two-minute incubation with the substrate over intracellular diphenhydramine without substrate addition, and SD is the corresponding standard deviation. MDAI ratio and SD are the corresponding parameters with 5,6-methylenedioxy-2-aminoindane. The OCT1 ratio is the ratio of cell uptake in OCT1 overexpressing cells over uptake in mock transfected cells. In the table, we differentiated between ratios below 0.7 with diphenhydramine antiport, possibly indicating good substrates; below 1.0, possibly indicating any substrates; and above 1.0 indicating non-substrates. With OCT1, any statistically significant ratio above 1.5 also indicates substrates of OCT1, whereas ratios above 3 may indicate good substrates.

**Table 5 ijms-23-08430-t005:** Analytical details on tandem mass-spectrometric detection.

	Q1 (Da)	Q3 (Da)	DP (V)	CE (V)	CXP (V)	Internal Standard
Used as probe substances						
Diphenhydramine	256.2	167.0 (152.0)	46	17 (49)	10 (10)	Tamoxifen
Imipramine	281.2	86.1 (58.1)	90	23 (59)	16 (10)	Duloxetine
MDAI	178.0	161.0 (131.0)	43	17 (27)	10 (16)	Ranitidine-d6
Memantine	180.2	163.1 (107.0)	46	21 (35)	10 (6)	Fenoterol-d6
**Analyzed as possible substrates**						
Amitriptyline	278.2	91.0 (117.1)	36	36 (30)	16 (8)	Amitriptyline-d6
Bupropion	240.1	183.9 (139.0)	55	17 (35)	12 (12)	Hydroxybupropion-d6
Citalopram	325.2	109.1 (262.2)	100	34 (27)	6 (18)	Doxepin
Clenbuterol	277.1	203.0 (258.9)	66	23 (15)	14 (18)	Caffeine
Clozapine	327.2	270.1 (192.2)	101	31 (57)	18 (12)	Fenoterol-d6
Desipramine	267.1	72.1 (44.2)	787	30 (62)	12 (8)	Duloxetine
Desmethylvenlafaxine	264.3	58.1 (107.2)	60	47 (50)	10 (6)	Desmethylvenlafaxine-d6
Doxepin	280.1	107.1 (77.0)	61	31 (75)	6 (14)	Escitalopram
Duloxetine	298.1	154.0 (44.1)	43	9 (31)	10 (8)	Imipramine
Ephedrine	166.0	148.1 (133.0)	41	17 (27)	9 (8)	Ranitidine-d6
Escitalopram	325.2	109.1 (262.2)	100	34 (27)	6 (18)	Doxepin
Fluoxetine	310.2	293.2 (148.1)	56	9 (13)	20 (10)	Duloxetine
Meptazinol	234.5	133.1 (107.0)	96	31 (35)	12 (20)	Raniditine-d6
Methylphenidat	234.2	84.1 (91.1)	71	27 (73)	6 (8)	Hydroxybupropion-d6
Metoclopramide	300.2	227.0 (183.9)	65	25 (42)	14 (12)	Milnacipran
Metoprolol	268.2	116.1 (74.0)	86	27 (35)	8 (14)	Fenoterol-d6
Morphine	286.2	201.1 (165.1)	110	36 (54)	10 (10)	Caffeine
N-Desmethyltramadol	250.2	232.3 (44.1)	55	12 (38)	15 (10)	Atropine
Nortriptyline	264.2	233.2 (91.1)	46	21 (30)	16 (6)	Amitriptyline-d6
Oxycodone	316.3	298.2 (256.2)	91	27 (35)	10 (8)	Raniditine-d6
Pyrazinamide	124.1	79.0 (107.1)	56	25 (15)	14 (20)	Ranitidine-d6
Sufentanil	387.2	238.1 (110.9)	81	27 (53)	8 (8)	Nalmefene
Urapidil	388.3	233.1 (190.0)	91	33 (47)	16 (12)	Nalmefene
Venlafaxine	278.2	121.0 (58.1)	65	39 (47)	15 (10)	Desmethylvenlafaxine-d6
**Internal standards**						
Amitriptyline-d6	284.3	91.0	71	34	7	-
Atropine	290.2	142.2 (124.2)	100	45 (33)	12 (12)	-
Caffeine	195.2	138.1 (110.0)	70	27 (32)	8 (8)	-
Desmethylvenlafaxine-d6	270.3	64.2 (107.0)	65	47 (48)	11 (6)	-
Fenoterol-d6	310.3	109.1 (141.0)	70	40 (26)	12 (12)	-
Hydroxybupropion-d6	262.2	244.0 (138.9)	61	17 (37)	16 (10)	-
Milnacipran	247.2	230.2 (100.1)	51	17 (27)	14 (8)	-
Nalmefene	340.3	322.2 (55.1)	61	29 (63)	10 (10)	-
Ranitidine-d6	321.0	176.0 (130.1)	65	25 (35)	15 (15)	-
Tamoxifen	372.2	71.9 (70.0)	90	47 (75)	14 (6)	-

Q1, mass selected at the first quadrupole; Q3, mass selected at the third quadrupole; DP declustering potential; CE collision energy; CXP, cell exit potential; numbers in brackets are qualifier masses and their detection settings.

## Data Availability

All relevant data are provided in the tables and figures.

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
