# Peer review of "Substrates of the Human Brain Proton-Organic Cation Antiporter and Comparison with Organic Cation Transporter 1 Activities"

_ijms, 2022, doi:10.3390/ijms23158430_

Round 1

Reviewer 1 Report

(1)  In Table 1, please provide sources in the table caption on where the authors obtain the information of all the drugs.

(2)  Also in Table 1, please provide full description of TPSA in the caption. Although the authors introduced this term in the manuscript, it was described after Table 3.

(3)  Can the authors rearrange drugs in Table 2 alphabetically, similar to Table 1?

(4)  Can the authors confirm that Figure 2 represents the statement in the manuscript: “…shown in Figure 2. As seen, for 24 of the 26 organic cations…”? Should there be 24 concentration-dependent uptake curves? If a drug was excluded, please indicate it in the caption.

Author Response

(1)  In Table 1, please provide sources in the table caption on where the authors obtain the information of all the drugs.

Yes, now we did so, this was in the methods section and is now also in the table caption.

(2)  Also in Table 1, please provide full description of TPSA in the caption. Although the authors introduced this term in the manuscript, it was described after Table 3.

Yes, the abbreviation is explained now with a few more words. It is also explained with one reference in results description of Figure 5.

(3)  Can the authors rearrange drugs in Table 2 alphabetically, similar to Table 1?

Yes, we did so

(4)  Can the authors confirm that Figure 2 represents the statement in the manuscript: “…shown in Figure 2. As seen, for 24 of the 26 organic cations…”? Should there be 24 concentration-dependent uptake curves? If a drug was excluded, please indicate it in the caption.

Yes, this was unfortunately handled not consequently. With sufentanil we could not make the 2nd and 3rd replicate (substance stock ran out and laborious German addictive drug laws).  Therefore, we firstly wanted to exclude sufentanil but since the curve looks ok, and trans-stimulation and cis-inhibition also was ok, we left it in and explained it in the caption to the figure.

Reviewer 2 Report

The article has an interesting and important topic for the medical field. Congratulations on your hard work.

Beside hCMEC/D3 cell line that has been used in this study, which represents endothelial cells, cells from the connective tissue, have you considered to use?   

Author Response

We thank the reviewer for the kind comments.

The comment concerning different cell types was, as we understand this, more a suggestion for other future research. We completely agree and we also got several cell lines, but that is a longer term project, also because several of these different cell lines possibly also expressing the H+OC antiporter all have their own requirements.